# Conformational Plasticity of Hepatitis B Core Protein Spikes Promotes Peptide Binding Independent of the Secretion Phenotype

**DOI:** 10.3390/microorganisms9050956

**Published:** 2021-04-29

**Authors:** Cihan Makbul, Vladimir Khayenko, Hans Michael Maric, Bettina Böttcher

**Affiliations:** 1Rudolf Virchow Center, Center for Integrative and Translational Bioimaging, University of Würzburg, 97080 Würzburg, Germany; cihan.makbul@uni-wuerzburg.de (C.M.); vladimir.khayenko@uni-wuerzburg.de (V.K.); hans.maric@virchow.uni-wuerzburg.de (H.M.M.); 2Biocenter, University of Würzburg, 97074 Würzburg, Germany

**Keywords:** hepatitis B core protein, hepatitis B virus, peptide inhibitor of envelopment, isothermal titration calorimetry, electron cryo microscopy, low-secretion phenotype mutants, peptide microarray

## Abstract

Hepatitis B virus is a major human pathogen, which forms enveloped virus particles. During viral maturation, membrane-bound hepatitis B surface proteins package hepatitis B core protein capsids. This process is intercepted by certain peptides with an “LLGRMKG” motif that binds to the capsids at the tips of dimeric spikes. With microcalorimetry, electron cryo microscopy and peptide microarray-based screens, we have characterized the structural and thermodynamic properties of peptide binding to hepatitis B core protein capsids with different secretion phenotypes. The peptide “GSLLGRMKGA” binds weakly to hepatitis B core protein capsids and mutant capsids with a premature (F97L) or low-secretion phenotype (L60V and P5T). With electron cryo microscopy, we provide novel structures for L60V and P5T and demonstrate that binding occurs at the tips of the spikes at the dimer interface, splaying the helices apart independent of the secretion phenotype. Peptide array screening identifies “SLLGRM” as the core binding motif. This shortened motif binds only to one of the two spikes in the asymmetric unit of the capsid and induces a much smaller conformational change. Altogether, these comprehensive studies suggest that the tips of the spikes act as an autonomous binding platform that is unaffected by mutations that affect secretion phenotypes.

## 1. Introduction

Hepatitis B virus (HBV) is a major human pathogen that belongs to the family of Hepadnaviridae. Up to now, 9 different genotypes of human HBV have been described [1]. In the pre-vaccination era, HBV-infected about a third of the world population [2,3]. The majority of the infected stage a full recovery, but some become chronic carriers with an increased risk for developing liver cirrhosis or primary liver cancer. Especially infants and young children are likely to grow into chronic carriers. Thus, worldwide vaccination programs are primarily targeting this population. Currently, there are around 250-million chronic carriers, most of them born before the infant vaccination program [4]. Chronic carriers rely on antivirals, which interfere with the viral maturation cycle to control their viral load. Some antivirals target the virus assembly and are now under clinical trials (for review, see [5,6]).

HBV is an enveloped virus, which consists of two shells. The inner shell is an icosahedral capsid formed by 240 copies of hepatitis B core protein (HBc) [7,8,9]. Depending on the genotype, HBc consists of either 183 or 185 amino acids. The *N*-terminal domain of HBc is mainly α-helical and assembles into dimers, which form protruding spikes at the capsid surface [10,11,12]. The *C*-terminal domain is largely disordered and contains several phosphorylation sites [13] and an arginine-rich domain (ARD) that together regulate genome interaction (reviewed in [14]). The capsid accommodates the viral polymerase and a viral, partially double-stranded DNA genome of 3.2 kb.

The capsid is surrounded by an outer shell that is formed by a lipid envelope densely packed with multiple copies of three types of surface proteins (HBs). These surface proteins have a common *C*-terminal region that comprises four transmembrane helices and represents the smallest of the surface protein (S-HBs). The two larger surface proteins are called medium HBs (M-HBs) and large HBs (L-HBs) and have additions of increasing length to the N-terminus.

During virus assembly, capsids form around the viral polymerase associated with a pre-genomic RNA (pgRNA) [15]. Inside the capsids, the viral polymerase transcribes the pgRNA into the mature, partly double-stranded DNA genome. Concomitantly, pgRNA is degraded [16]. It is likely that during this process, nucleotides enter and leave the capsid, probably through the large holes at the base of the spikes. Capsids with a mature DNA genome, as well as empty capsids, are enveloped and secreted, while capsids-containing RNA or single-stranded DNA are not enveloped [17,18,19,20].

It is still controversial how HBs in the envelope and HBc in the capsid interact. The formation of virions requires S-HBs and L-HBs, while M-HBs are dispensable. In enveloped capsids, the tips of the spikes of the capsids touch the envelope [8,9]. This suggests that the tips of the spikes could provide a binding platform for the interaction with HBs. However, mutational screens showed that none of the amino acids at the tips of the spikes is essential for the formation of mature virions [21]. Instead, the same screen identified several residues, L60, K96 and L95, close to the entrance of a hydrophobic pocket in the center of the spikes as being important for virion formation and the interaction with L-HBs [22]. Some naturally occurring mutations at this site is also implicated with either low-level secretion (P5T-HBc and L60V-HBc; [23]) or premature (F/I/97 L-HBc; [24,25]) secretion phenotypes. It was suggested that such mutations might change core-envelope recognition [21], particularly as point-mutations of L-HBs can offset the phenotype [26].

Peptides based on the hexapeptide “LLGRMK” block the interaction between capsids and L-HBs and inhibit virion formation [27,28]. These peptides bind to the tips of the spikes where envelope and capsid are in touch [8,9] but are distant from the hydrophobic pocket. Here we revisit two peptides of this family (P1: “MHRSLLGRMKGA”; P2: “GSLLGRMKGA”) and clarify whether mutations in HBc that change the secretion phenotype also impact the binding of these peptides.

## 2. Materials and Methods

### 2.1. Purification of HBc-CLPs

The purification of HBc-CLPs (genotype D; strain ayw; GenBank: V01460.1. [29]) was performed similarly as outlined before for split-cores [30] with modifications: BL21 (DE3) Star cells were transformed with pRSF-T7-HBcOpt-wt, pRSF-T7-HBcOpt-P5T, pRSF-T7-HBcOpt-F97L and pRSF-T7-HBcOpt-L60V, respectively. These DNA constructs were generated by site-directed mutagenesis of pRSF-T7-HBcOpt-wt. The cells were grown in a terrific broth medium (with 34 µg/mL chloramphenicol and 50 µg/mL kanamycin) at 37 °C under vigorous shaking to an OD of ca. 5. Then the temperature was reduced to 24 °C and after 30 min the induction was started with 0.5 mM IPTG for 16 h. The cells were harvested at 4 °C by centrifuging at 4000× *g* for 30 min. 80 g cell pellet was transferred into a beaker and topped up to 400 mL (20% *w/v*) with TN300 buffer (100 mM Tris-HCl, 300 mM NaCl, 5 mM MgCl_2_, 5 mM CaCl_2_, 6 mM 2-mercaptoethanol, 2 mM PMSF and 2 mM benzamidine, pH 7.5). The cells were disrupted at 1500 bar by microfluidization, and the lysate was cleared by centrifugation at 8000× *g* for 90 min at 4 °C. The supernatant was precipitated by slowly adding of a 100% saturated ammonium sulfate solution (pH 7.5 saturated at 20 °C) to a final saturation of 40% (*w/v*) under gentle stirring in an ice/water bath. The precipitate was recovered by centrifugation at 2000× *g* and 30 min at 4° and washed with 500 mL TN300 buffer containing ammonium sulfate (40% final saturation, *w/v*) and 6 mM 2-mercaptoethanol. The suspension was recovered by centrifugation at 2000× *g* for 30 min at 4°, and the pellet was dissolved in 400 mL TN300 buffer with 2 mM 2-mercaptoethanol. The solution was precipitated with ammonium sulfate (30% final saturation, *w/v*) under gentle stirring in an ice/water bath. The precipitate was recovered by centrifugation at 2000× *g* and 30 min at 4 °C. The pellets were solubilized 1:5 (*v/v*) in TN300 buffer with 5 mM 2-mercaptoethanol and subjected to density-gradient centrifugation (steps with 10, 20, 30, 40, 50 and 60% sucrose (*w/v*) in TN300 buffer) in an SW 32 Ti rotor (Beckman Coulter GmbH, Krefeld, Germany) at 4 °C and 125,000× *g* for 4 h. Each sucrose gradient step and the sample had a volume of 5.5 mL. Subsequently, the topmost 16.5 mL (which includes the sample volume and the 10 and 20% sucrose gradient steps) were discarded, and 12 fractions (each 1 mL) were collected. The concentrations of the fractions were determined by the Bradford assay. The protein concentrations of fractions 1 and 2 were too small for determination. Therefore, only fractions 3–12 (which corresponds to 30–40% sucrose) were analyzed, first by native agarose gel electrophoresis (NAGE) as described with the difference that nucleic acids were stained by the SYBR Safe dye (1:30,000 dilution, *v/v*). Of each fraction, 10 µg protein was applied onto the agarose gel. NAGE was performed at 60 V for 90 min with 1% agarose in 2 × TAE buffer. The agarose gels were re-stained with Coomassie brilliant blue as well. The same 10 fractions were subjected to SDS–PAGE (15% acrylamide) and stained with Coomassie brilliant blue. Unlike for the NAGE analysis, only 5 µg of protein per fraction was applied onto the acrylamide gel. Fractions with high purity were pooled, dialyzed, and their concentrations were determined by the Bradford assay [31]. 3 µg of each protein was applied to the acrylamide gel.

### 2.2. Automated Solid-Phase Peptide Synthesis

μSPOT peptide arrays (doi.org:10.1002/qsar.200640130) were synthesized using a MultiPep RSi robot (CEM GmbH, Kamp-Lintford, Germany) on in-house produced, acid labile, amino-functionalized, cellulose membrane discs containing 9-fluorenylmethyloxycarbonyl-β-alanine (Fmoc-β-Ala) linkers (average loading: 130 nmol/disc—4 mm diameter). Synthesis was initiated by Fmoc deprotection using 20% piperidine (pip) in dimethylformamide (DMF) followed by washing with DMF and ethanol (EtOH). Peptide chain elongation was achieved using a coupling solution consisting of preactivated amino acids (aas, 0.5 M) with ethyl 2-cyano-2 (hydroxyimino) acetate (oxyma, 1 M) and *N*,*N’*-diisopropylcarbodiimide (DIC, 1 M) in DMF (1:1:1, aa:oxyma:DIC). Couplings were carried out for 3 × 30 min, followed by capping (4% acetic anhydride in DMF) and washes with DMF and EtOH. Synthesis was finalized by deprotection with 20% pip in DMF (2 × 4 μL/disc for 10 min each), followed by washing with DMF and EtOH. Dried discs were transferred to 96 deep-well blocks and treated, while shaking, with side-chain deprotection solution, consisting of 90% trifluoracetic acid (TFA), 2% dichloromethane (DCM), 5% H_2_O and 3% triisopropylsilane (TIPS) (150 μL/well) for 1.5 h at room temperature (rt). Afterward, the deprotection solution was removed, and the discs were solubilized overnight (ON) at rt, while shaking, using a solvation mixture containing 88.5% TFA, 4% trifluoromethanesulfonic acid (TFMSA), 5% H_2_O and 2.5% TIPS (250 μL/well). The resulting peptide-cellulose conjugates (PCCs) were precipitated with ice-cold ether (0.7 mL/well) and spun down at 2000× *g* for 10 min at 4 °C, followed by two additional washes of the formed pellet with ice-cold ether. The resulting pellets were dissolved in DMSO (250 μL/well) to give final stocks. PCC solutions were mixed 2:1 with saline–sodium citrate (SSC) buffer (150 mM NaCl, 15 mM trisodium citrate, pH 7.0) and transferred to a 384-well plate. For transfer of the PCC solutions to white-coated CelluSpot blank slides (76 × 26 mm, Intavis AG Peptide Services GmbH and CO. KG), a SlideSpotter (ICEM GmbH) was used. After completion of the printing procedure, slides were left to dry ON.

### 2.3. Peptide Microarray-Binding Assay

The microarray slides were blocked for 60 min in 5% (*w/v*) skimmed milk powder (Carl Roth) phosphate-buffered saline (PBS; 137 mM NaCl, 2.7 mM KCl, 10 mM Na_2_HPO_4_, 1.8 mM KH_2_PO_4_, pH 7.4). After blocking, the slides were incubated for 30 min with 55 nM (monomer equivalent) of HBc in the blocking buffer, then washed 3× with PBS. HBc was detected with a primary 1:2500 diluted mAb16988 (anti-hepatitis B virus antibody, core antigen, clone C1-5, aa 74-89, MilliporeSigma, Darmstadt, Germany) and a secondary 1:5000 diluted HRP-coupled Anti-mouse antibody (31,430, Invitrogen). The antibodies were applied in blocking buffer for 30 min, with three PBS washes between the antibodies and after applying the secondary antibody. The chemiluminescent readout was obtained using SuperSignal West Femto maximum sensitive substrate (Thermo Scientific GmbH, Schwerte, Germany) with a c400 Azure imaging system (lowest sensitivity, 10 s exposure time).

Binding intensities were quantified with FIJI [32] using the “microarray profile” plugin (OptiNav Inc, Bellevue, WA, USA). The raw grayscale intensities for each position were obtained for the left and right sides of the internal duplicate on each microarray slide, *n* = 4 arrays in total. Blank spots were used to determine the average background grayscale value that was subtracted from the raw grayscale intensities of non-blank spots. Afterward, the spot intensities were normalized to the average grayscale value of the 15 replicates of peptide binder P2 (“GSLLGRMKGA”).

### 2.4. Semipreparative Peptide Synthesis

Standard solid-phase peptide synthesis with Fmoc chemistry was applied, shortly: TentaGel rink-amide resin (0.25 mmol/g, INTAVIS Peptide Services GmbH & Co. KG, Tübingen, Germany) was swollen in DMF, then deprotected using 20% Piperidine solution in DMF, and after washes, the first amino acid (4 eq.) was coupled to the resin using oxyma (4 eq.) and DIC (4 eq.). Further deprotection and conjugation cycles were performed using the same reagents. Coupling efficiency was monitored by measuring the absorption of the dibenzofulvene–piperidine adduct after deprotection. Peptide side-chain deprotection and cleavage from the resin were done using a cocktail of 90.5% TFA, 4% H_2_O, 3% TIPS, 2.5% dithiothreitol for 2 h, with agitation at rt. The peptides were precipitated in ice-cold ether, purified with HPLC, and analyzed by liquid chromatography–mass spectrometry (LC–MS, Appendix A).

### 2.5. Isothermal Titration Calorimetry (ITC)

The peptides P1 (“MHRSLLGRMKGA”) and P2 (“GSLLGRMKGA”) are soluble up to 8 mM. For ITC experiments with P1, peptide and sample were equilibrated against ITC buffer 1 (20 mM HEPES, 50 mM NaCl, 1 mM MgCl_2_, 1 mM CaCl_2_, pH 7.5). For all other peptides, peptides and samples were equilibrated against ITC buffer 2 (40 mM HEPES, 200 mM NaCl, 1 mM MgCl_2_, 1 mM CaCl_2_, pH 7.5). The respective ITC buffer was prepared, filtered (220 nm pore size) and degassed. The dialysis membrane tubing (Spectra/Por Biotech cellulose ester tubing, 1000 kDa MWCO, Spectrum Laboratories, Inc., Rancho Dominguez, CA, USA) was washed with the ITC buffer before use. Fractions from the density gradient with intact HBc capsids were pooled (4–10 mL), filtered by a 220 nm pore size filter (Rotilabo syringe filter, Carl Roth GmbH + Co. KG, Karlsruhe, Germany) and transferred into the dialysis membrane tubing. Samples were dialyzed against 1.4 L ITC buffer under gentle stirring at 4 °C overnight for 16 h. The next day, the dialysate was concentrated in a concentrator (30 kDa MWCO Spin-X UF 6 mL, Corning Inc., Corning, NY, USA), filtered by a centrifugal filter unit with a pore size of 0.1 µm (Ultrafree MC, VV 0.1 µm, Merck KGaA, Darmstadt, Germany). The peptides were dissolved in the dialysis buffer at the end of the dialysis. The protein concentrations were determined with the Bradford assay [31].

Before carrying out ITC experiments, peptide and HBc solutions were degassed for 10 min at 20 °C (ThermoVac, Malvern Panalytical, Malvern, Worcestershire, UK). The ITC experiments were performed using a MicroCal iTC200 instrument (Malvern Panalytical, Malvern, Worcestershire, UK), and thermodynamic parameters were calculated with the origin-based software provided by the instrument supplier. For more details of the ITC conditions, see Appendix A.

Samples of HBc and P2 dilutions used in the ITC experiments were retrieved and stored at −20 °C. The concentrations of these samples were determined by the ninhydrin assay [33] and compared with values obtained by the Bradford assay (HBc concentrations) and gravimetry (peptide concentrations). If there were major deviations in the concentrations determined by the Bradford assay and gravimetry, the values from the ninhydrin assay were used to refit the ITC isotherms.

### 2.6. Differential Scanning Calorimetry (DSC)

The differential scanning calorimetry experiments were conducted with a Nano DSC instrument (TA Instruments, New Castle, DE, USA) with a scan rate of 2 °C/min between 20 and 125 °C. HBc samples were prepared for the ITC experiments. For each HBc construct, at least 2 independent scans were performed at concentrations between 0.75 and 6 g/L in ITC buffer 2.

### 2.7. Negative-Staining Electron Microscopy (EM)

Homemade carbon-coated copper grids were rendered hydrophilic by glow discharging in the air at a pressure of 29 Pa for 1 min at medium-power using a plasma cleaner (model PDC-002. Harrick Plasma Ithaca, NY, USA). 3.5 µL of each sample was applied onto the grids and incubated for 1 min. Excess sample from the grids was removed by blotting, followed by washing twice with water, twice with 2% uranyl acetate. Then the grids were stained with a 2% uranyl acetate solution for 5 min. After removing excess uranyl acetate solution by blotting, the grids were air-dried. The grids were imaged (Tecnai T12 TEM, FEI Company, Hillsboro, OR, USA) at an acceleration voltage of 120 kV and a nominal magnification of 42,000.

### 2.8. Electron Cryo Microscopy (Cryo-EM)

Fractions confirmed for the presence of intact and pure HBc-CLPs were subjected to a buffer exchange using concentrators with a molecular weight cutoff of 100 kDa. To this end, the protein concentrate was five-fold diluted by TN20 buffer (20 mM Tris-HCl pH 7.5, 50 mM NaCl, 1 mM MgCl_2_ and 1 mM CaCl_2_) and concentrated by centrifugation at 4000× *g* and 4 °C. To the concentrate, fresh TN20 buffer was added, and the centrifugation repeated several times until the sucrose concentration dropped below 0.2% (*v/v*). The concentrations of the protein concentrates were determined with the Bradford assay. For imaging capsids with bound peptide, the peptide was dissolved in TN20 buffer and added to the capsid concentrate in a 10-fold molar excess final relative to the HBc monomer concentration (0.2–0.25 mM HBc monomer vs. 2–2.5 mM peptide). All samples were filtered once by centrifugal filter units with a pore size of 100 nm (Ultrafree MC, VV 0.1 µm, Merck KGaA, Darmstadt, Germany). Alternatively, samples were used at the endpoint of the ITC experiment without further modification.

Grids coated with holey carbon (copper grids, 300 mesh, R 1.2/1.3. Quantifoil Micro Tools, Jena, Germany) were rendered hydrophilic by plasma discharging in the air at a pressure of 29 Pa for 2 min at medium-power in a plasma cleaner (model PDC-002. Harrick Plasma Ithaca, NY, USA). 3.5 µL of each sample was pipetted onto the grids and plunge frozen in liquid ethane by a Vitrobot mark IV (FEI Company, Hillsboro, OR, USA). The Vitrobot settings were: Blot force of 25, wait time of 0 s, drain time of 0 s, blot time of 6 s, 100% humidity, 4 °C and filter paper Whatman 541. All vitrified samples were stored for at least one night in liquid nitrogen before loading into the electron microscope.

Movies of vitrified capsids were recorded with a Titan Krios G3 electron microscope (Thermo Fisher Scientific, FEI Deutschland GmbH, Dreieich, Germany) on a Falcon III direct detector in integrating mode as previously described [34]. The nominal magnification was 75,000 with a calibrated pixel size of 1.064 Å. The total exposure varied between different samples and was between 30 e/Å² and 80 e/Å² (Appendix A), with an exposure rate of 18 e/(px² s).

### 2.9. Image Processing

Movie frames were motion-corrected, dose-weighted and averaged in 5 × 5 patches with “motioncorr2” [35]. For HBc-L60V, capsids were selected with “cryolo” [36] using the general neuronal network. All subsequent processing was done with Relion 3.1 [37]. In brief: Particles from a subset of the “cryolo”-selected particles (14,600 particles from 228 micrographs) were extracted (“relion_preprocess”) with a box-size of 440 × 440 Px² and downsampled to 128 × 128 Px². Extracted particles were aligned and classified, and two of the 2D class averages were chosen as templates for auto-picking of all subsequent datasets. Auto-picked particle images were extracted (440 × 440 Px²) and downsampled to 220 × 220 Px² or 128 × 128 Px² followed by 2D classification. Particle images grouping into classes representing intact, centered T = 4 capsids were re-extracted with 440 × 440 Px² followed by auto-refinement (“relion_refine”) with a previously determined map of F97L [34] as start reference and imposing icosahedral symmetry. After refinement, the data were 3D-classified into 3–5 classes without further alignment. Particles from the best resolved, class were selected for CTF-refinement, including beam tilt correction and per-particle defocus refinement. Afterward, particles orientations were locally refined (“relion_refine”). The resolution was estimated by Fourier shell correlation between half-maps after gold-standard refinement (“relion_postprocess”) and corrected for the contribution of the mask (Supplemental Information S2).

For some data sets, we analyzed the variations in the asymmetric unit of the capsids by asymmetric processing: After the last symmetric refinement, the particles were symmetry expanded (“relion_particles_symmetry_expand”). This generated 60 copies per capsid, with orientations that placed a copy of every asymmetric unit at the same position. After symmetry expansion, the asymmetric units were repositioned to place the local three-fold symmetry axis in the center of the box and re-extracted with a box size of 128 × 128 Px². From the re-extracted particles of F97L + P1, we calculated a 3D map (“relion_reconstruct”). This map was used as starting reference for the subsequent 3D classification. In addition, we generated a 3D mask that loosely covered the two dimers in the asymmetric unit. For this, the current model of the asymmetric unit of F97L + P1 was fitted into the 3D map using Chimera [38]. The densities with a radial distance of less than 4 Å to the model coordinates were extracted with Chimera “vop zone”. From the extracted densities, a mask is generated with “relion_mask_create”. Extracted particle images were 3D-classified without further alignment using the mask to focus the classification. For most data sets, two rounds of 3D classification were carried out. For the first classification, the regularization parameter T was between 5 and 10, and 5 to 20 classes were calculated. In this classification, typically, some class averages had lower maximal densities or lower resolution and were excluded from further processing. In the subsequent 3D classification, the remaining data were classified into 5 classes with a regularization parameter of T = 15 and without alignment. To estimate the positional variability between classes, the classes were translationally aligned with the “fit in map” tool of Chimera, optimizing the correlation between class averages.

For comparisons, the maps were truncated at the same resolution and scaled to give the same radial intensity distribution in their power spectra. For this, the map of P5T + P1 was chosen as a reference map after B-factor sharpening and low-pass filtering to 3.5 Å resolution. The other maps were scaled with “relion_image_handler” to give the same intensity profile of the power-spectra as that of the reference map. These scaled maps were used for calculating difference maps or local correlation maps.

### 2.10. Model Building, Refinement, and Validation

For the model building of P5T-HBc, L60V-HBc and P1/P2 bound to WT-HBc, P5T-HBc, F97L-HBc and L60V-HBc, the pdb model 6HTX [39] was modified and fit into EM maps. The resulting models were real-spaced, refined with Phenix [40,41], and validated [42]. All figures depicting EM maps and the corresponding PDB models were generated with Chimera [38].

## 3. Results

We purified P5T-HBc, L60V-HBc, WT-HBc and F97L-HBc by ammonium sulfate precipitation followed by a sucrose density step gradient. All four HBc variants migrated in the same fractions in the gradient and formed capsids with packaged RNA as confirmed by native agarose gel electrophoresis (NAGE) stained with SYBR safe (nucleic acid stain) and Coomassie brilliant blue (protein stain). For further analysis, we pooled the fractions, which showed a single band on NAGE (Figure 1) and confirmed the homogeneity of capsids by electron microscopy of negatively stained samples (Appendix A).

All four HBc variants had formed capsids with a homogeneous size distribution corresponding to a T = 4 assembly with 240 subunits per capsid [10]. The thermal stability of the HBc variants was assessed by differential scanning calorimetry (DSC). All four HBc variants showed heat flux curves with a single peak indicating that the complete capsid melts simultaneously and cooperatively. L60V-HBc was the most stable variant with the highest transition temperature T_m_ (T_m_(L60V-HBc) > T_m_(WT-HBc) > T_m_(F97L) > T_m_(P5T)) (Figure 1). While L60V-HBc, WT-HBc and F97L-HBc had similar transition temperatures between 91 °C and 95 °C, the transition temperature of P5T-HBc was much lower (T_M_ = 86 °C), showing that the mutation P5T weakens the stability of the whole capsid substantially.

### 3.1. Peptide-Binding Affinity Is Affected by Mutations in the Center of the Spike

Next, we tested whether the HBc capsids could bind P2 and determined the thermodynamic binding properties by isothermal titration calorimetry (ITC). The binding of P2 was exothermic for all four variants (Table 1, Figure 2). Fitting of the isotherms gave a stoichiometry of *n* = 0.5 for P5T-HBc, WT-HBc and F97L-HBc, suggesting that each HBc dimer in the capsid bound one peptide. For L60V-HBc, we could not determine a meaningful stoichiometry (Figure 2d) and fixed the stoichiometry to *n* = 0.5 for determining the K_D_ and ∆H. The stoichiometry of *n* = 0.5 was also justified by the subsequent analysis of the electron microscopic data (see below) that showed one peptide per dimeric spike in L60V-HBc. The K_D_s were similar for WT-HBc and P5T-HBc (K_D_WT_ = 68 ± 4 µM, K_D_P5 T_ = 74 ± 5 µM), but according to Student’s *t*-test significantly larger for L60V-HBc and F97L-HBc (K_D_L60V_ = 127 ± 19 µM, and 155 ± 14 µM).

### 3.2. The Low-Level Secretion Phenotype Mutation L60V, but Not the Mutation P5T Mobilizes the F97 Side-Chain in the Center of the Spikes

For elucidating the binding mechanism of P2 to HBc capsids, we compared capsids with and without bound peptides with the same genetic background (genotype D; strain ayw; GenBank: V01460.1.). EM maps and pdb models of WT-HBc and F97L-HBc without bound peptides have been determined previously [34,39]. In addition, we determined a map of WT-HBc with our electron microscope camera setup to generate maps for the comparisons with the same magnification and information transfer as all the other maps in our study. The previous model of WT-HBc (6HTX, [39]) was refined against this map and showed essentially the same structure but resolved fewer residues at the C-terminus.

Structures of the low-secretion variants HBc-P5T and HBc-L60V [23] have not been reported previously and were determined for the first time in this study. Both variants formed icosahedral capsids with protruding spikes and adjacent holes. The overall resolutions were 3.2 Å with B-factors of 139 Å² (P5T-HBc) and 158 Å² (L60V-HBc), respectively (Appendix A). Generally, the protein shells were better resolved than the protruding spikes. The maps of P5T-HBc and L60V-HBc were almost indistinguishable from WT maps (local cross-correlation > 0.9) except at regions close to the mutated residues (local correlation < 0.9, Figure 3). Here, L60V-HBc showed somewhat larger differences, which affected the spikes surrounding the 5-fold symmetry axes (AB spikes) more than those surrounding the 3-fold symmetry axes (CD spikes).

Next, we built models into the maps of L60V-HBc and P5T-HBc. These models had the same backbone conformation as WT-HBc confirmed by RMSD values between the Cα atoms of less than 1 Å (RMSD_P5T_ = 0.5 Å, RMSD_L60V_ = 0.7 Å). The per-residue B-factors of both models were higher at the tips of the spikes than in the continuous protein shell, which is also expected from the variation in local resolution of the respective maps (Appendix A). Similar observations were previously reported for HBc-F97L and HBc-WT [39] and suggested that the spikes are more mobile than the rest of the capsids. However, this local variation did not completely explain the overall high B-factors of the maps of HBc variants in this and other studies [11,39,43,44,45].

To understand the reason for the high overall B-factors, we analyzed the conformational and positional variability of the asymmetric units within the capsids by 3D classification without alignment. The analysis showed that the asymmetric units were randomly displaced by up to 2 Å in respect to their symmetry-related positions (Appendix A). Models of the asymmetric units refined against the maps of the different classes confirmed the relative displacements between the Cα atoms of up to 2.3 Å. After rigid-body alignment of the individual chains, the RMSD decreased to 0.4 Å, which is consistent with the expected variability of models built at this resolution [46]. Thus, the high overall B-factors of the capsids are mainly caused by rigid-body displacements of the individual chains rather than significant conformational changes within the chains.

One noticeable difference between the consensus models was at the position of F97, which is spatially close to P5T and L60V. While in WT-HBc and P5T-HBc, the phenylalanine side-chain of F97 pointed towards the dimer axis, in L60V-HBc, it had an additional alternate position pointing away from the dimer axis (Figure 3g). Based on the density observed in the EM-map, almost 50% of all F97 side chains must have adopted this alternate conformation. The alternate orientation of F97 in L60V-HBc increases the adjacent hydrophobic pocket similar to the smaller side-chain in F97L-HBc does, which also points away from the dimer axis [39] (Figure 3g).

### 3.3. Binding of Peptides Splays the Tips of the Spikes

Next, we evaluated whether mutations that lead to different secretion phenotypes impact how peptides bind to the tips of the spikes. Therefore, we determined structures for WT-HBc, F97L-HBc, P5T-HBc and L60V-HBc with bound peptides (Appendix A). Peptides were added in 10-fold molar excess to obtain full occupancy of the binding sites. P1 and P2 peptides bound to the tips of the spikes at the dimer interface and altered the whole protruding part of the spikes as evidenced by the low local correlation between maps of the same variant with and without bound peptides (Figure 4 and Appendix A).

Model-building showed that peptide-binding splayed the loops at the tips of the spikes apart (Figure 5). The splaying was transmitted throughout the protruding part of the spike tilting the ascending helix α3 and the descending helix α4 outwards. Tilting pivoted at residues G63 in α3 and G94 in α4, respectively. The rearrangements in the upper spike region resulted in an RMSD of 2.5 Å in the position of Cα atoms (residues 63–94) between HBc dimers with and without bound peptides. Model building of HBc into the maps leaves unaccounted density at the tips of the spikes between the rearranged loops that were attributed to the peptide density (Figure 4 arrow). The length of this density could accommodate 5–6 amino acids, which is only a fraction of the P1 and P2 peptides. The elongated flat shape of the density matches an extended stretch, but not an α-helix. The binding site for the peptides is at the dimer interface at a local twofold-symmetry axis. Therefore, the peptide could bind in the two symmetry-related orientations, which would lead to an average of both possible orientations as suggested by the two-fold symmetric shape of the density (Figure 4b).

Peptide binding induced the same structural changes of the spike region independent of the HBc variant to which the peptide was bound. This is evidenced by the high local cross-correlation (>0.95) at the binding site between different HBc variants with bound peptides (Figure 4 and Appendix A). Therefore, we conclude that the binding mechanism of the peptides is structurally conserved and independent of point mutations related to the different secretion phenotypes.

Often binding of a ligand leads to more rigid structures than in the unbound form. Here, we noticed that the per-residue B-factors at the tips of the spikes remained high in the peptide-bound state, suggesting no effect of peptide binding onto the intrinsic flexibility of the spikes. To decide whether such flexibility is due to incomplete occupancies or other factors, we analyzed WT-HBc with bound P2 by classification of the asymmetric unit (Appendix A). All class averages showed density for the bound peptide and the splaying of the helices in the upper spike region without detectable variation in the occupancy. The dimers had a similar positional variability as the dimers in HBc capsids without bound peptides but also additional conformational variability in the upper spike region. The variability was largest for chain C in the descending helix α4 with an RMSD between Cα atoms (residues 79–94) of 2.9 Å (Figure 5c–e). Changes in the ascending helix α3 of chain C (residues 61–79) were smaller and had an RMSD of 2 Å in the most diverging classes. In contrast, the continuous protein shell (residues 1–61 and 94–142) had RMSDs of less than 0.3 Å, confirming the conformational rigidity of this part. All other chains were less variable in the upper spike region (RMSD between Cα of residues 61–94 of the most divergent classes: chain A: 0.6 Å, chain B: 1.4 Å, chain C: 2.4 Å and chain D: 1.0 Å). Thus, the four chains within the asymmetric unit varied in their conformational flexibility much more than without bound peptides. Classification of P5T-HBc + P2, L60V-HBc + P2 and F97L-HBc + P2 confirmed a similar variability of the spikes.

### 3.4. The Core-Binding Motif of the Peptide Is “SLLGRM.”

Our structural studies showed that P1 and P2 bound to the dimer interface at the tips of the spikes, as reported earlier [27]. Unfortunately, the conformational variability at the tips of the spikes, together with possible binding of the peptides in two alternate orientations, did not reveal the exact residues of the peptide at the binding interface. Earlier experiments suggested that the negatively charged residues E77 and D78 at the tips of the spikes in HBc, if mutated to alanine, greatly reduce the binding of P2 [27]. Thus, it was speculated that the two positively charged residues in P2 with the methionine in the center could be important for binding. This notion was further supported by phage display experiments, which showed weakened binding if methionine in the center of the motif is exchanged against other hydrophobic amino acids [28]. Therefore, we speculated that the minimal binding peptide might be “GRMKG”, which is symmetric in its distribution of positively charged residues around the methionine. However, structure determination by electron cryo microscopy (Appendix A) showed no association of “GRMKG” with the capsids and no conformational rearrangements that are indicative for binding. Additionally, ITC experiments confirmed no measurable affinity between WT-HBc and this shortened peptide (Appendix A, Table 1), consistent with earlier observations in phage display competition assays [28].

To identify the core binding motif, we characterized the importance of each residue of P2 for binding to HBc by µSPOT peptide microarrays [47]. Probing *C*-terminal and *N*-terminal truncated peptide variants revealed “SLLGRM” as the core-binding motif (Figure 6a). A full positional screen probing all possible point mutants of “GSLLGRMKGA” determined the exact sequence requirements for the peptide recognition, namely S, L, L, G, X, ψ (ψ = hydrophobic) (Figure 6).

To validate the array-based experiments with the *C*-terminally immobilized peptides, we determined the exact binding affinities of three different *N*-terminally unmodified and terminally amidated peptides in solution using ITC (Appendix A, Table 1). As expected, “SLLGRM” maintained binding towards WT-HBc, albeit with reduced affinity (130 µM compared to 68 µM for the elongated peptide). Mutation within the strictly conserved core motif abolished binding (no binding detected for “SLKGRM” and “SLLGEM”).

Next, we determined the structure of WT-HBc with the bound core binder “SLLGRM” at the end of the ITC experiment. This ensured that more than 95% of the binding sites were occupied with “SLLGRM”. The map showed that only the CD spikes bound “SLLGRM” but not the AB spikes (Figure 7). The density in the cleft between the two loops accommodated some 5 residues at the same position as identified for P1 and P2 binding. This confirmed that “SLLGRM” is the core-binding motif and that it is bound with its central “LG”-part to the dimer axis. Surprisingly, binding of “SLLGRM” induced much smaller conformational changes in the CD spikes than the binding of P2 (Figure 7) and P1. This suggested that distant residues, which are only present in the longer constructs, support the rearrangement of the spike region. The structural changes in the CD spike accumulated in the upper part of the descending helix α4 (residues 79–84) in chains C and D. This part was poorly resolved, indicating high mobility or even unwinding of the helix. In the center of the CD spike, F97 in chain D adopted two alternate rotamer conformations (Figure 7b) similar as observed for L60V-HBc without bound peptide (Figure 3).

## 4. Discussion

In this study, we present the first structures of capsids of the HBc mutants P5T-HBc and L60V-HBc, with a low-secretion phenotype [23]. Although both mutants share the same phenotype, they show very different thermal stability, with P5T-HBc having a 9 K lower transition temperature (T_m_) as L60V-HBc. At a resolution of 3.2 Å, the mutants P5T-HBc and L60V-HBc have the same overall structure as WT-HBc without rearrangements in the protein’s backbone. The positional mobility of the dimers within the capsids around their symmetry-related positions is 2–3 Å as predicted from all-atoms molecular dynamic (MD) simulations of *C*-terminally truncated WT capsids [48]. Thus, our structural analysis provides experimental evidence for the simulated structural dynamics of capsids and confirms that these predictions are also valid for full-length variants with randomly packaged host–RNA.

The two mutations of the low-secretion phenotype mutants, P5T and L60V, are at the entrance of a hydrophobic pocket in the center of the spikes. This site is a hotspot for secretion modulating mutations [21,23,24,26]. While L60V is part of the spike, P5T is in the N-terminus at the dimer interface and occludes the entrance to the pocket. 5 T/P and 60 V/L are in direct contact with each other but are located on opposite chains, thereby contributing to the dimer interface. Our detailed analysis of this region shows only small changes in the side-chain densities of these two residues compared to WT-HBc. However, local cross-correlation maps and DSC indicate that the mutations P5T and L60V impact differently on the capsid properties: While the mutation P5T weakens the stability of the whole capsids (lowest T_M_), but leaves the spikes unchanged (high local correlation), L60V stabilizes the capsid slightly (largest T_M_) and changes the spike around the pocket (lower local correlation). As the mutations L60V and P5T have opposite effects on the stability, it is unlikely that changes in their direct interaction are causative for the low-secretion phenotypes.

All four HBc variants bind the peptides P1 and P2 at the tips of the spikes in the groove between the two loops that delineate the dimer interface. In our hands, the shorter P2 is better soluble and induces the same conformational changes as the longer P1, which binds stronger, but causes aggregation of HBc capsids. Therefore, we focused our comparative binding studies on P2. All four HBc variants bind P2 (“GSLLGRMKGA”) weakly with K_D_s between 68 µM and 160 µM (Table 1). Compared to the WT-HBc and P5T-HBc, which bind P2 with intermediate micromolar affinity (68 µM and 74 µM), the premature (F97L-HBc) and low secretion (L60V-HBc) variants bind P2 much weaker (160 µM and 127 µM). Thus, the binding affinity is correlated with changes in the center of the spike but not with the secretion phenotype.

The two stronger binders (P5T-HBc and WT-HBc) have essentially the same spike structure, whereas, in the two weaker binders (L60V-HBc and F97L-HBc), a large hydrophobic side chain is exchanged for a smaller hydrophobic side chain, increasing the size of the adjacent pocket. Although these mutations are within the invariant scaffold of the capsid, they are spatially close to the pivot points at which the upper spike region tilts to accommodate the peptide binding. The pivot points G63 and G94 and the mutation L60V and F97L are direct neighbors in adjacent turns of the helix. In addition, the side chain of F97 in L60V-HBc can adopt two alternate positions. One of them increases the size of the adjacent pocket similar to in F97L-HBc for the smaller Leu side chain [39]. Therefore, it is conceivable that the side-chain mobility of residue 97 affects the conformational adaptability at the distant peptide-binding site. In this context, it is interesting to note that WT-HBc with bound “SLLGRM” also exhibits mobility of the F97 side-chain (Figure 7) and has a similar low affinity for this shorter binder as F97L-HBc and L60V-HBc have for the longer P2.

We observe dissociation constants for P2 ranging widely between 68 µM to 160 µM depending on single point mutations far away from the actual binding site. The binding is weak, and most of the bound peptides can be easily removed from the capsids by washing, as evidenced by EM maps of washed capsids (not shown). Our tightest binder for P2 is WT-HBc (K_D_ = 68 µM) and has a similar affinity as reported earlier for the longer P1 peptide “MHRSLLGRMKGA” bound to *C*-terminally truncated HBc (K_D_ = 78 µM) [49]. However, a much lower dissociation constant (K_D_ = 6.8 µM) with a much stronger heat signature was reported for P1 in another study [50]. In our hands, P1 also bound somewhat stronger to WT-HBc (K_D_ = 26 µM) than P2, but not with such a strong heat signature. The apparent differences in binding constants may be related to the genetic backgrounds in which the studies were carried out. Crystallization experiments of truncated HBc variants suggested that distinct changes in crystallization properties relate to the genotype [51]. In this earlier study, the variants were referred to as HBc∆-CW (diffracted to 3.3 Å, 1QGT [52]), HBc∆-Riga (diffracted to 23 Å), and HBc∆-Edin (did not diffract). The best diffracting variant, HBc∆-CW, was also used in the study where the tightest binding was observed (K_D_ = 6.8 µM [50]). The non-diffracting variant, HBc∆-Edin, showed the weakest binding (K_D_ = 78 µM [49]) and was used in the phage display-based discovery and initial characterization of “LLGRMK” peptides [28].

The construct in our study is genotype D; strain ayw; GenBank: V01460.1. [29]) and is closely related to its naturally occurring variant HBc-Riga. Both variants differ only in positions, T33N and A80I. HBc-Riga was also used in previous EM studies [10,11] for mapping the peptide-binding site of P2 [27]. In HBc-Riga, P2 was bound to the tips of the spikes but did not induce conformational changes in the spike. Similarly, we observe no conformational changes upon binding the shorter core peptide “SLLGRM” to our variant (Figure 4f,g). The binding site of “SLLGRM” is delineated by A80 at either site, which is mutated to isoleucine in HBc-Riga. This suggests that residue 80 contributes to the binding-induced conformational switching.

The differences between our ayw strain and the much tighter-binding CW-variant are four residues (V74N; S87N, F97I, I116L), which do not contact the peptide directly. However, two of the residues, V74N and S87N, are in the ascending helix α3 and in the descending helix α4 of the upper spike, which moves upon peptide binding. NMR studies on the truncated CW dimers identify 87 N as one of the residues with a change in the chemical environment, further highlighting its involvement in peptide binding [50]. The other residue, 74, is much closer to the peptide-binding site but cannot be assigned in the bound and unbound state due to its mobility in CW dimers. The same NMR study describes a whole network of interacting residues in the upper and central spike regions, changing their chemical environment upon peptide binding. This network agrees with the rearrangements in the capsid that we observe for P2 binding, but not with the lack of changes for binding the shorter “SLLGRM”.

P2-binding intercepts the intra-dimer interface and leads to a more open conformation at the tips of the spikes. This open conformation is also adopted in the absence of binding partners by certain mutants Y132A-HBc [53] and D78S-HBc [54] that affect HBc-assembly. While Y132A-HBc forms planar trimers of dimers but no capsids, D78S-HBc assembles slowly into stable capsids but has weakened dimer stability [54]. MD-simulations show that the mutation D78S destabilizes the descending helix α4 [54], where we see the largest mobility upon P2-binding (Appendix A). Thus, binding of P2 changes the conformation of the HBc spike similarly as mutations do that affect the capsid assembly and lead to splaying of the loops in the absence of binding partners. In this respect, the mutations D78A-HBc or E77A-HBc that reduce binding of P2-related peptides [27] must be evaluated in a new light. These mutations probably induce a conformational change at the tips of the spikes that similarly weakens the dimer interface as D78S rather than being involved in the direct interaction with P2.

The core binding motif of P2 is “SLLGRM”, which is sandwiched at the dimer interface between E77-D78 on either side. In capsids, “SLLGRM” can only bind to the CD spikes, which are more flexible than the AB spikes [48]. Binding probably disrupts the intra-dimer interaction similarly as described for D78S-HBc in the absence of a binding partner. Consistent with the disruption of the dimer interface, “SLLGRM”-binding destabilizes the upper part of the helix α4 (weak density) but cannot induce the full conformational change seen for P1 and P2. Thus, an additional contribution from amino acids in the longer P2 is required to stabilize the full conformational change that leads to the enhanced binding of P2 and P1.

“LLGRMK” related peptides interfere with the association between HBs and HBc capsids in competition assays [28]. Therefore, it is likely that HBs and peptides compete for the same binding site. Docking of NMR structures of binding fragments derived from L-HBs and S-HBs suggests that they bind to the tips of the spikes in the open conformation with the splayed loops [53,55] as P2 does. The pocket in the center of the spike is too small to accommodate peptide binders or the binding site of HBs directly. Its importance is probably related to fine-tuning the conformational plasticity of the upper spike, which is required for a binding competent state at the tips of the spikes. Whether such a binding competent state is directly relevant for the viral maturation in vivo remains to be shown.

In conclusion, our investigations show that the tips of the spikes provide a conformational switch that can be addressed by binding effector molecules. We think that this is valuable information for targeting the conformational plasticity of the spikes. Such effectors would act on the assembled capsids before envelopment in contrast to the established capsid assembly modulators [56]. This is a still unexplored mode of action for HBV antivirals [6], and we believe that exploiting this potential may lead to developing new therapeutics.

## Figures and Tables

**Figure 1 microorganisms-09-00956-f001:**
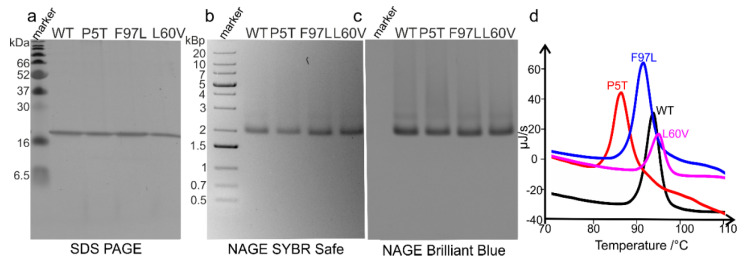
Purification and thermal stability of different HBc variants. (**a**) Purification products of recombinant WT-HBc (genotype D; strain ayw; GenBank: V01460.1.) and its mutants P5T-HBc, F97L-HBc and L60V-HBc were analyzed by 15% SDS–PAGE stained with Coomassie brilliant blue. For each construct, 3 µg protein was applied. The molecular weights of the protein standards in the first lane are indicated in kDa. The following lanes from left to right contain purified WT-HBc, P5T-HBc, F97L-HBc and L60V-HBc; (**b**,**c**) show native agarose gel electrophoresis (NAGE) of 7 µg of the same constructs as in (**a**) on a 1% agarose gel. The first lane shows the DNA-marker with the size of the DNA standards indicated in kilo base-pairs; (**b**) NAGE stained with SYBR Safe, which stains nucleic acids; (**c**) NAGE stained with Coomassie brilliant blue, which stains protein, but not nucleic acids. Electron micrographs of the corresponding negatively stained HBc variants are shown in Appendix A; (**d**) differential scanning calorimetry (DSC) of the four HBc variants. The plot shows the heat flux plotted over the temperature. The transition temperatures for the capsids of the four constructs are: T_M_(P5T-HBc) = 86.2 °C, T_M_(F97L-HBc) = 91.3 °C, T_M_(WT-HBc) = 93.7 °C and T_M_(L60V-HBc) = 94.9 °C.

**Figure 2 microorganisms-09-00956-f002:**
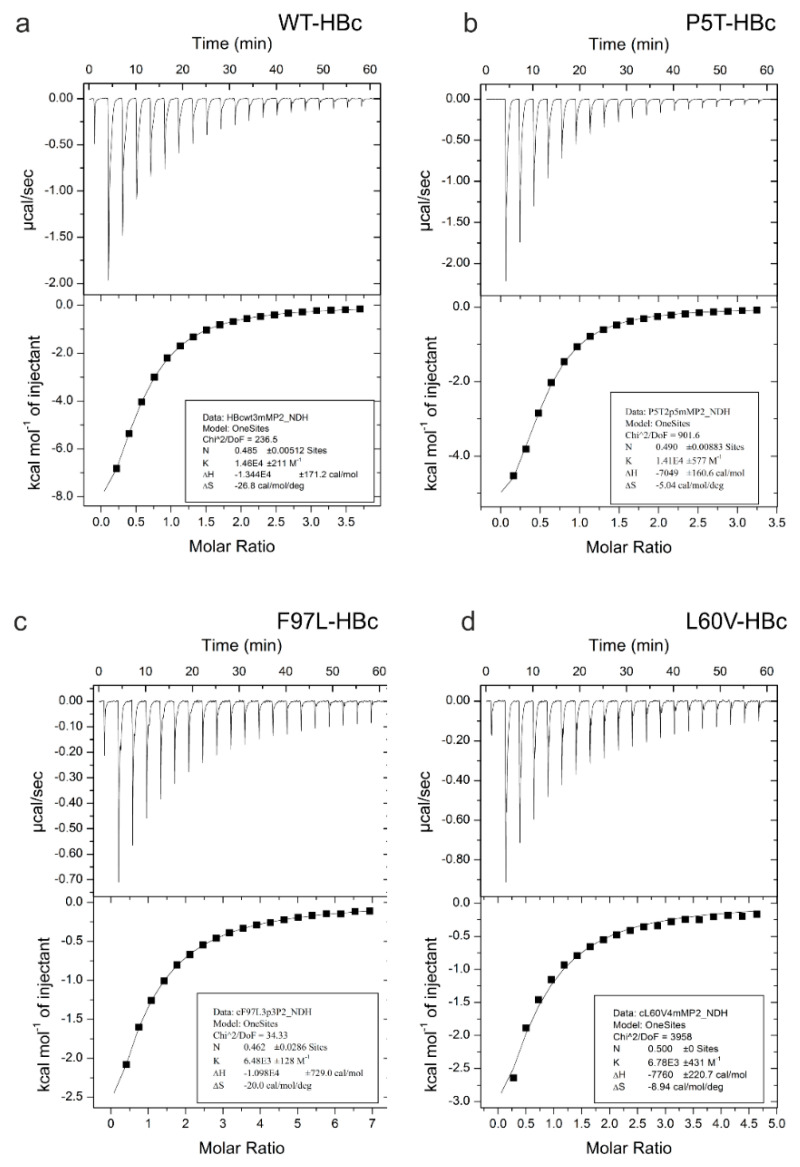
ITC of WT-HBc (**a**), P5T-HBc (**b**), F97L-HBc (**c**) and L60V-HBc (**d**). The plots (**a**–**d**) show a characteristic titration experiment between capsids of HBc variants in the cell and peptide P2 in the syringe. The upper panel shows the raw ITC isotherms in power versus time. Each peak corresponds to an injection of P2. The area under each peak is proportional to the heat produced at each injection. The lower binding curve shows the total heat developed per mol P2 in dependence on the molar ratio of P2/HBc_monomer_. The fitted binding curve is shown as a solid line. The fitting results for a single-binding site model are listed in the insert. The fitted parameters were n, ∆H and K_A_ = 1/K_D_ in panels (**a**–**c**) and ∆H and K_A_ = 1/K_D_ with a fixed *n* = 0.5 in panel (**d**).

**Figure 3 microorganisms-09-00956-f003:**
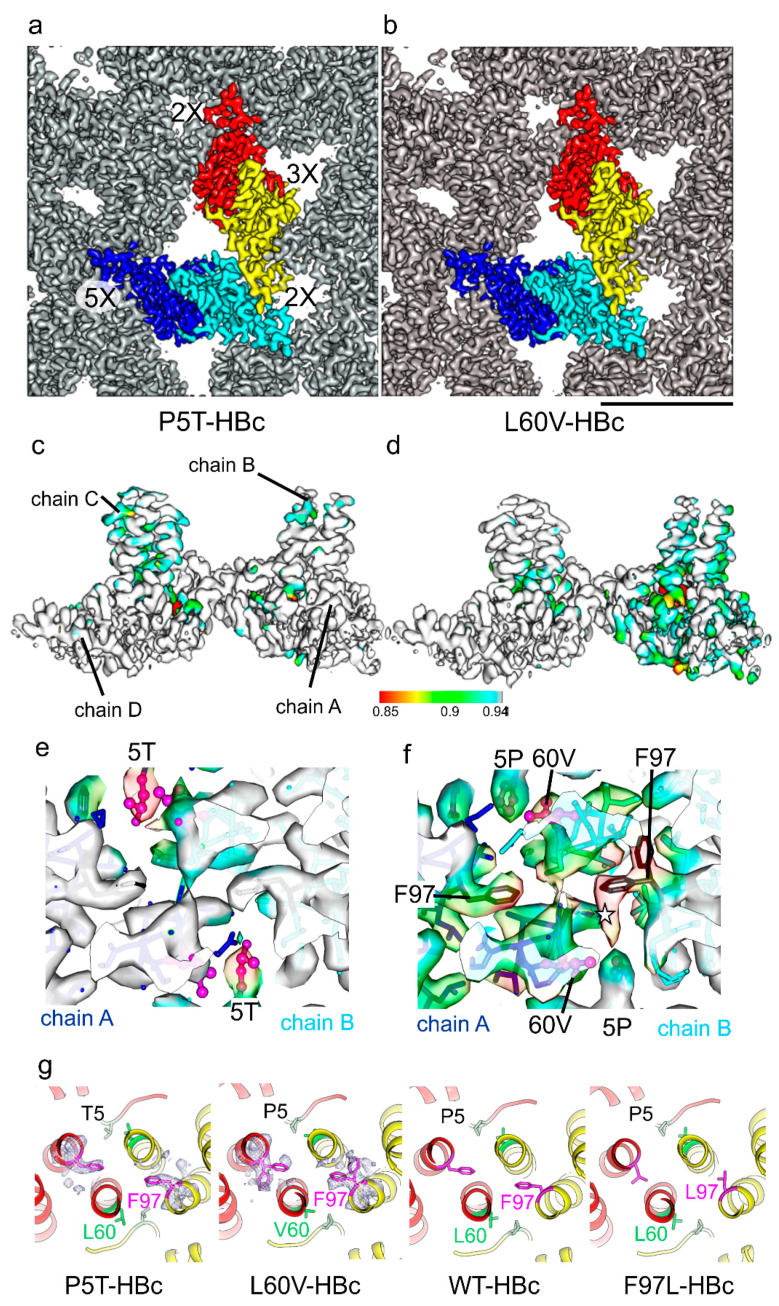
Structure determination of the low secretion phenotype mutants P5T-HBc (**a**,**c**,**e**) and L60V-HBc (**b**,**d**,**f**). Close-up of the capsid surface of P5T-HBc (**a**) and L60V-HBc (**b**). One asymmetric unit of the icosahedral capsid is colored with different colors as follows: chain A: blue; chain B cyan, chain C yellow and chain D red. The nearest 2-fold, 3-fold and 5-fold symmetry axes are indicated in (**a**) The scale bar in (**b**) indicates 5 nm and applies for panels (**a**,**b**); (**c**,**d**) show the maps for an asymmetric unit colored with its local correlation with the map of WT-HBc determined in this study. All maps are filtered to a resolution of 3.5 Å and scaled to give the same radial intensity distribution of the power spectra. The color key for the local correlation is given below. Its length corresponds to 5 nm and applies for (**c**,**d**); (**e**,**f**) show a 1 nm thick slice through the center of the AB spike where the local correlation is lowest in L60V-HBc. The maps are fitted with the respective models. Areas with lower correlation are also more transparent. Residues 5 and 60 are shown in magenta; residue 97 is shown in black. In P5T-HBc, the local correlation is lowest at the point mutation but leaves the spike unaffected. In L60V-HBc, the low local correlation identifies an unaccounted density (star) and two alternate orientations of F97; (**g**) shows a comparison of the models of P5T-HBc, L60V-HBc, WT-HBc and F97L-HBc (6HU4 [39]) at the center of the CD spike viewed along the dimer axis. Chain C is colored in yellow and chain D in red. The side chains of residues 5, 60 (green) and 97(magenta) are shown in all four models. The mutation L60V destabilizes the rotamer orientation of F97 that adopts two alternate positions in L60V-HBc. The density around F97 is shown as mesh in P5T-HBc and L60V-HBc.

**Figure 4 microorganisms-09-00956-f004:**
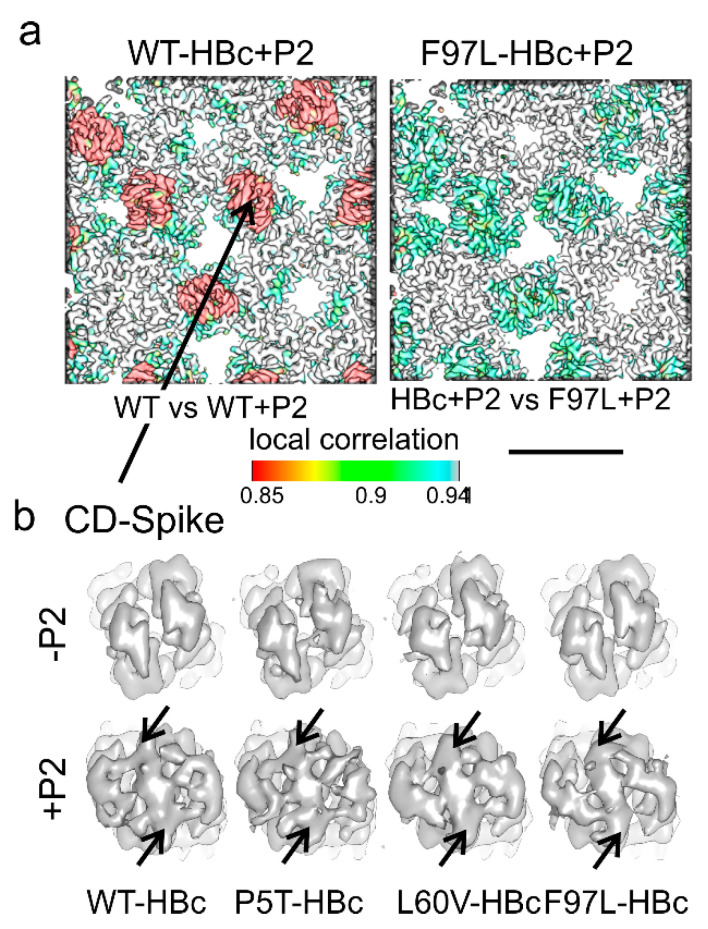
Capsid surface of HBc variants with bound peptide: (**a**) Two exemplary representations of the capsid surface close to a local 3-fold symmetry axis are shown for WT-HBc with bound P2 on the left and for F97L-HBc with bound P2 on the right. WT-HBc binds P2 strongest and F97L-HBc weakest. For WT-HBc, the surface is colored by the local correlation between WT-HBc-maps with and without bound P2. The local correlation is low in the upper half of the spikes (red), highlighting differences between the maps attributed to peptide binding and the induced conformational changes. For F97L-HBc, the surface is colored by the local correlation between F97L-HBc + P2 and WT-HBc + P2. The local correlation is high at all places suggesting that the HBc variants adapt similarly to P2 binding. The color key for the local correlation is given below and is the same as in Figure 3. The scale bar indicates 5 nm. A table with all maps colored by local correlation is shown in Appendix A. (**b**) Close-up of the tip of the CD spike without bound peptide (upper row) and with bound P2 bottom row. The map of F97L-HBc is taken from the EMDB entry 4417 [34] and scaled in the radial intensity profile of the power spectrum to match the other maps. All other maps were determined in this study. The density attributed to P2 is marked at either end by arrows. Broadening of the spikes with bound P2 is apparent.

**Figure 5 microorganisms-09-00956-f005:**
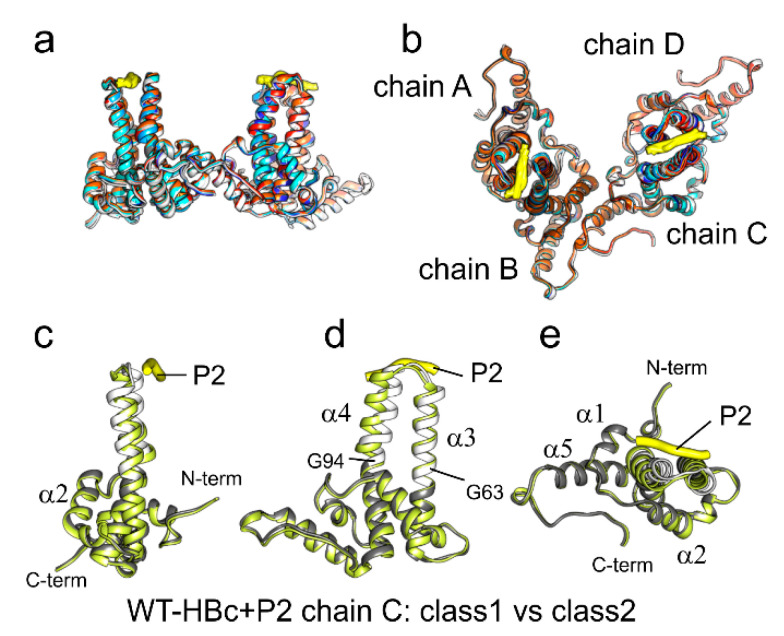
Models of HBc variants with bound peptide. (**a**,**b**) models of the asymmetric units with bound P1 or P2 are superposed. The peptides are shown in yellow, and the chains of the HBc variants are colored as follows: L60V-HBc + P2 in cyan, P5T-HBc+P1 in blue, P5T-HBc + P2 in dark blue, WT-HBc + P1 in white, WT-HBc + P2 in gray, F97L-HBc + P2 in red and F97L-HBc + P1 in orange. All models show essentially the same fold. The views in (**a**,**b**) are related by a 90° rotation around the horizontal axis; (**b**) shows a view from the outside of the capsids towards the center; (**c**–**e**) show three perpendicular views of the models of chain C (yellow and white/gray) modeled into the two most diverging classes of the asymmetric unit of WT-HBc + P2 (Appendix A). The two models were aligned by rigid-body alignment to compensate for their relative displacement. The two models diverge in the upper spike region hinged at G94 and G63. Most asymmetric units are similar to the yellow model, while the white/gray model represents only 15–20% of the asymmetric units in the analysis.

**Figure 6 microorganisms-09-00956-f006:**
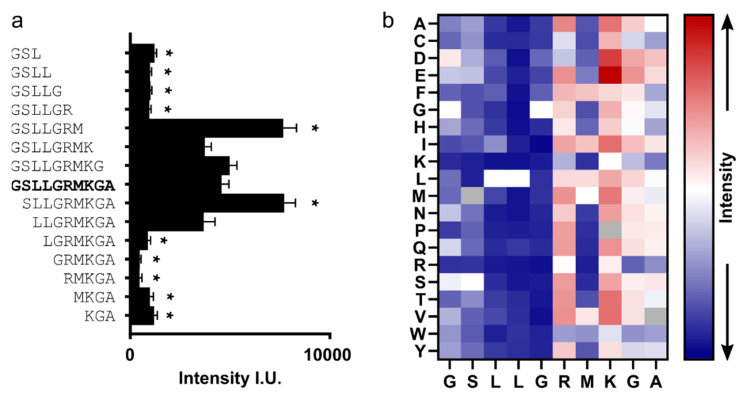
Relative binding strength of P2-permutations and terminal truncations to WT-HBc. (**a**) Chemiluminescent-based intensity readout of WT-HBc binding to *N*-terminally or *C*-terminally truncated P2 variants. Higher intensity correlates to the stronger binding. Mean + SEM. Sequences marked with a star differ significantly from the no truncated sequence (“GSLLGRMKGA”). Significance determined using ANOVA test with a follow-up Dunnett’s test for multiple comparisons. *p* < 0.0001. *n* = 12; (**b**) heat map of positional point mutation microarray screen of P2 to WT-HBc. Shown are intensity values of point-mutated variants normalized to the corresponding WT P2 sequence (“GSLLGRMKGA”). Higher spot intensity (red) correlates with stronger binding and vice versa (blue). Gray color indicates n/a. In the horizontal direction, the positions of P2 are shown; in the vertical direction, the respective point mutation is indicated. Values are presented as the mean of *n* = 4. The raw chemiluminescent readout data are shown in Appendix A.

**Figure 7 microorganisms-09-00956-f007:**
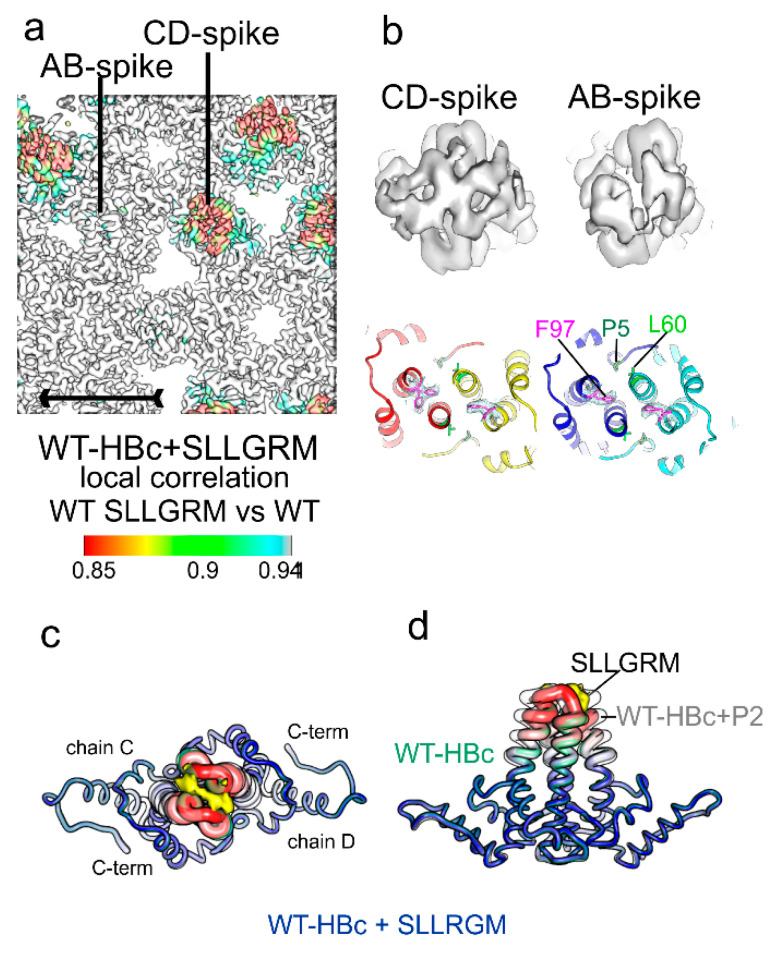
Map and Model of WT-HBc + “SLLGRM”: (**a**) close-up of capsid surface of WT-HBc with bound “SLLGRM” and colored with the local correlation between the maps of WT-HBc +/− “SLLGRM” (Appendix A). The tips of the CD spikes, but not the AB spikes, differ as indicated by the lower local correlation and seen in the close-ups of the spikes in the upper panel in (**b**); the lower panel in (**b**) shows the models at the center of the spikes with the side chains of P5, L60 and F97 shown in dark green, light green and magenta, respectively. The map density around F97 is shown in mesh representation. In the CD spike, F97 adopts to alternate orientation in chain D; (**c**,**d**) show chains C and D of WT-HBc + “SLLGRM” superposed with the models of WT-HBc (light green, transparent) and WT-HBc + P2 (white, transparent). The density attributed in the map to “SLLGRM” is shown as surface representation in yellow. WT-HBc + ”SLLGRM” is colored according to its per-residue B-factors (low blue B-factors, high red B-factors). The radius of the tubes in the models relates to the mean displacement calculated from the B-factors. WT-HBc + ”SLLGRM” is more like WT-HBc without bound peptide (light green transparent) than to WT-HBc + P2 (white transparent).

**Table 1 microorganisms-09-00956-t001:** Thermodynamic characterization of the interaction between P2, P1, “SLLGRM”, “SLLGEM”, “SLKGRM” and “GRMKG” with different HBc variants.

HBc-Variant	Peptide	Nr. of Repeats	K_D/_(µM)	Ratio Peptide/HBc	∆H/(kcal/mol)	∆ST/(kcal/mol)
WT-HBc	P2	6	68 ± 4	0.48 ± 0.06	−14.4 ± 1.6	−8.8 ± 1.6
P5T-HBc	P2	6	74 ± 5	0.45 ± 0.05	−7.7 ± 0.8	−2.1 ± 0.9
F97L-HBc	P2	5	155 ± 14	0.52 ± 0.07	−9.9 ± 1.0	−4.7 ± 0.9
L60V-HBc ^1)^	P2	3	127 ± 19	Fixed: 0.5	−6.6 ± 1.2	−1.4 ± 1.3
WT-HBc	P1 *	1	26	0.45 ± 0.02	−9.0 ± 0.6	−2.5
WT-HBc	“GRMKG”	1	dnb			
WT-HBc ^1)^	“SLLGRM”	1	130 ± 7	Fixed: 0.25	−8.3 ± 0.2	−3.2
WT-HBc	“SLLGEM”	1	dnb			
WT-HBc	“SLKGRM”	1	dnb			

Binding of P1 * was determined at 37 °C in ITC buffer 1 and at 20 °C in ITC buffer 2 for all other peptides. “GRMKG”, “SLLGEM” and “SLKGRM” did not bind HBc (dnb). For experiments with only one repeat, the fit errors are given; otherwise, the standard deviations are shown. ^1^ For some experiments, affinities were too low to determine a meaningful ratio of peptide to HBc; in these cases, the stoichiometry was fixed based on the number of observed binding sites in the EM maps and only ∆H and K_D_ were determined by curve-fitting. The titration experiment and the fitted isotherm for the experiments with one repeat are shown in Appendix A.

## Data Availability

Cryo-EM maps reported in this article have been deposited in the Electron Microscopy Data Bank (EMDB) with ID-codes: EMD-12810 (L60V-HBc), EMD-12815 (P5T-HBc), EMD-12819 (WT-HBc), EMD-12820 (WT-HBc+P2), EMD-12821(WT-HBc + SLLGRM), EMD-12822 (L60V-HBc + P2), EMD-12823 (F97L-HBc+P1), EMD-12824 (F97L-HBc + P2) and EMD-12825 (P5T-HBc + P2). The molecular models were deposited in the Protein Data Bank (PDB) with ID codes: 7OCO (L60V-HBc), 7OCW (P5T-HBc), 7OD4 (WT-HBc), 7OD6 (WT-HBc + P2), 7OD7 (WT+ SLLGRM), 7OD8 (L60V-HBc + P2).

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
