# Peer review of "Conformational Plasticity of Hepatitis B Core Protein Spikes Promotes Peptide Binding Independent of the Secretion Phenotype"

_microorganisms, 2021, doi:10.3390/microorganisms9050956_

Round 1

Reviewer 1 Report

In this study, Makbul et al. studied the conformational plasticity of HBc protein spike. The authors did comprehensive structural studies and concluded the HBc tip provides the platform for peptide binding which is independent of the secretion phenotype. Overall, this study is informative and provides new insights on HBV capsid-envelope interactions. I have minor suggestions on their introduction part.

Line 38 -39: “Some of these antivirals target the virus assembly (for review see [5,6]).” is not an accurate statement because none of the antiviral target viral assembly has been approved. Suggest changing it as “Some of the antivirals target the virus assembly is under clinical trials”.

Line 46-47: “The capsid accommodates the viral polymerase and a viral partly double-stranded DNA genome of 3.2 kb, which is in contact with the ARD. “is not accurate. There is no evidence of the DNA genome in contact with ARD. Only known is about HBc CTD is responsible for viral genome (pgRNA) packaging.”

Line 57-58: “For this, nucleotides enter and leave the capsid through large holes at the base of the spikes.” is a reasonable hypothesis but lacks evidence. Please rephrase it.

Author Response

Reviewer 1: Line 38 -39: “Some of these antivirals target the virus assembly (for review see [5,6]).” is not an accurate statement because none of the antiviral target viral assembly has been approved. Suggest changing it as “Some of the antivirals target the virus assembly is under clinical trials”.

Response: We have changed our statement to "Some antivirals target the virus assembly and are now under clinical trials"

Reviewer 1: Line 46-47: “The capsid accommodates the viral polymerase and a viral partly double-stranded DNA genome of 3.2 kb, which is in contact with the ARD. “is not accurate. There is no evidence of the DNA genome in contact with ARD. Only known is about HBc CTD is responsible for viral genome (pgRNA) packaging.”

Response: We have deleted ", which is in contact with the ARD"

Reviewer 1: Line 57-58: “For this, nucleotides enter and leave the capsid through large holes at the base of the spikes.” is a reasonable hypothesis but lacks evidence. Please rephrase it.

Response: We appreciate the lack of evidence. Therefore, we have rephrased the sentence as follows: "It is likely that during this process nucleotides enter and leave the capsid probably through the large holes at the base of the spikes."

Reviewer 2 Report

April 17, 2012

Review for Microorganism.

“Conformational plasticity of Hepatitis B core protein spikes promotes peptide binding independent of the secretion phenotype.

Comment.

In the current enthusiasm among the investigators searching for the HBV cure drugs, the authors have performed commendable investigations and observed the plasticity of the conformational activity of the HB core protein spikes. The tips of the HBc protein spikes provided a conformational switch that can be addressed by binding of the effector molecules.

Peptides ‘LLGRMK’ are known to block the interaction between capsids and L-HBs and inhibit virion formation. These peptides bind to the tips of the spikes where envelope and capsid contact. ‘LLGRMK’ related peptides interfere with the association between HBs and HBc-capsid in competition assays.

Authors suggested that it is important to target the conformational plasticity of the spikes. Such effectors would act on the assembled capsids. These plasticity modulators would take action after capsids have formed and before they are enveloped. 

Based on their study, authors propose another future HBV cure area by targeting another step of viral maturation. 

Author Response

Response: Thank you for the review. No concerns have been raised. There is no change to the manuscript in response to this review. 

Reviewer 3 Report

This paper on the first structures of capsids of the HBc-mutants P5T-HBc 
and L60V-HBc, with a low secretion phenotype, is significant. The authors stated clearly what the study found and how they did it.

The title is informative and relevant. The references are relevant and recent. Appropriate and key studies are included.

The introduction reveals what is already known about this topic.

The variables are well defined and measured appropriately. The study methods are valid and reliable, a great variety of methods and techniques were used. There are enough details provided in order to replicate the study.

The data is presented in an appropriate way. The text in the results add to the data and it is not repetitive. Statistically significant results are clear. Results are discussed from different angles and placed into context without being overinterpreted.

The conclusions answer the aim of the study. The conclusions are supported by references and own results.

Specific comments on weaknesses of the article and what could be improved:

Major points  

  1. Please, state the limitations of the study
  2. Which results are with practical meaning and how could they be implemented in clinical laboratory practice?

Minor points

  1. There are some typos, such as a space between the number and the percentage (i.e., 30 %);

Author Response

Major points:

  1. Please, state the limitations of the study

Response: This is a biophysical/structural study, which dissects binding of a peptide to HBc-capsids of different variants. We do not know whether the observed conformational plasticity of the spike  and the conformational cross talk upon binding of peptides is relevant for virus maturation in vivo. To clarify this limitation we have added: "Whether such a binding competent state is directly relevant for the viral maturation in vivo remains to be shown."  (see discussion page 19).

  2. "Which results are with practical meaning and how could they be implemented in clinical laboratory practice?"

Response: Our basic biophysical and structural studies look into how binding of certain peptides changes the structure of the capsids.

None of our findings have immediate implications on clinical laboratory praxis. However, we think that there might be potential in developing plasticity modulators to interfere with viral maturation in future. These modulators do not necessarily need to bind at the tips of the spikes. It is probably more important for such molecules to arrest the spike in a binding incompetent state. 

We have rephrased our conclusion as follows:

" We believe that plasticity modulators would act after capsids have formed and before capsids are enveloped.  This is a still unexplored mode of action for HBV antivirals [6], and we believe that exploiting this potential may lead to the development of new therapeutics"

Minor points:

There are some typos, such as a space between the number and the percentage (i.e., 30 %);

Response: Spaces between numbers and "%" were deleted throughout the manuscript. Many  typos have been corrected throughout the text.